# Biological and Agrotechnical Aspects of Weed Control in the Cultivation of Early Potato Cultivars under Cover

**Piotr Pszczółkowski** [1],*[ID]**, Piotr Barbaś** [2]**, Barbara Sawicka** [3][ID]** and Barbara Krochmal-Marczak** [4]

[1] Experimental Station for Cultivar Assessment of Central Crop Research Centre, Uhnin, 21-211 Dębowa Kłoda, Poland

[2] Department of Potato Agronomy, IHAR-PIB, Branch in Jadwisin, Szaniawskiego 15, Str., 05-140 Serock, Poland; p.barbas@ihar.edu.pl

[3] Department of Plant Production Technology and Commodity Science, University of Life Sciences in Lublin Akademicka 15, str., 20-950 Lublin, Poland; barbara.sawicka@up.lublin.pl

[4] Department of Plant Production and Food Safety, Carpathian State University in Krosno, Dmochowskiego, 12, 38-400 Krosno, Poland; barbara.marczak@kpu.krosno.pl

\* Correspondence: p.pszczolkowski.inspektor@coboru.pl

**Abstract:** Problems with weed infestation under cover were the reason to conduct research on the regulation of weed infestation in potato cultivation for early harvest. The field experiment was carried out in 2015–2017 at the Experimental Station for Cultivar Assessment in Uhnin (51°34′ N, 23°02′ E) using the method of random subblocks, in a dependent system (split-split-plot). The first order factor was edible potato cultivars 'Denar' and 'Lord'. The second order factor was cultivation technologies: (A) traditional technology, (B) technology using polyethylene film cover, (C) technology using polypropylene agrotextile. The third order factor was weed management methods: (1) mechanical, (2) mechanical and chemical method using Afalon Dispersion 450 SC preparation, (3) mechanical and chemical methods using Racer 250 EC herbicide, and (4) mechanical and chemical methods using a mixture of herbicides Afalon Dispersion 450 SC and Command 480 EC. Mechanical and chemical methods proved to be more effective than the mechanical method. The best effectiveness in limiting both fresh and dry weed mass in potato cultivation under cover was achieved using the mechanical and chemical method using a mixture of herbicides, Afalon Dispersion 450 SC and Command 480 EC.

**Keywords:** early potato production; polypropylene agrotextile; polyethylene sheeting; herbicides; mechanical weed control; chemical weed control; potato cultivars; weed infestation

## 1. Introduction

The cultivation of very early potato cultivars under cover for a very early harvest requires the plants to be provided with appropriate thermal and humidity conditions due to the risk of spring frosts. The unfavorable influence of low temperatures in the initial period of potato growth increases the effectiveness of using foil and agrotextile covers [1–3]. Temperature of the soil under the agrotextile at a depth of 5 cm is 1–2 °C and at a depth of 10 cm it is 2–3 °C higher than that of uncoated soil [4]. Covers are placed flat, immediately after planting, and it is a procedure that accelerates the harvest of tubers and reduces the risk of plant freezing. The most popular covers used in early potato plantations include perforated foil, polypropylene non-woven fabric, and a biofilm that decomposes in contact with the soil [2–4].

Weed infestation is one of the factors determining the potato yield. It is favored by a wide row spacing, a long period from planting to plant emergence, slow initial growth of potato plants,

introducing manure combined with intensive mineral fertilization [5–8]. In an integrated cultivation system, chemical weed control is the basic treatment that eliminates segetal vegetation from the field of a cultivated plant [8,9].

Potato yield losses due to weed infestation in Europe are estimated at 10% to 70%. Damage resulting from limiting access to light, water, and nutrients should be added to them, as well as the host role of weeds relating to diseases and pests, harvest difficulties, increase in mechanical damage to tubers, and deterioration of their quality [10–12]. Scalla [13], Ackley et al. [14], Azadbakht et al. [10] proved that during the action of herbicides on weeds, a selection process takes place, due to which the plants that are the least sensitive to the preparation survive, and each weed population is more or less heterogeneous, and the resistance production process may vary in intensity and speed. Physiological and biochemical studies indicate differences between herbicide-resistant and herbicide-sensitive weeds, which are based on the rate of herbicide uptake and decomposition in tissues and different distribution of the root system in the soil (selectivity of uptake) [13,15,16]. Their distribution in cells and tissues also plays an important role. In plants resistant to a given herbicide, they are not displaced, or they migrate in small amounts to the tips of growth, which are very sensitive to toxic substances. The dominant role in the selectivity of action of these compounds is played by the plant's ability to detoxify them [15–17]. In the opinion of Hess and Foy [18] and Starck [16], the phenomenon of weed resistance to herbicides will increase. It is already much more serious than it is supposed, and the survival of weeds in plantations after treatment is most often attributed to inaccurate treatment or low-quality preparation.

Chemical control of weeds should be carried out in their early development stages [7,9,10]. Barbaś [5] and Nowacki et al. [9] recommend three dates of herbicide application: after potato planting (up to 10 days), a few days before the expected emergence of the plants, and after emergence, when the plants develop 2–4 leaves and reach a height of 10–15 cm. It is difficult to implement these recommendations in the cultivation of plants under cover, so the problem of controlling segetal vegetation in this cultivation technology still awaits a solution. Hence, the aim of the research was to develop a technology of growing very early potato cultivars under cover put on "flat" with the use of herbicides or herbicide mixtures, compared to the mechanical method, and the impact of this technology on the degree of weed infestation, species composition, and fresh and air-dried mass of weeds.

It was assumed in the research hypothesis that it is possible to cultivate very early potato cultivars with the use of polyethylene film covers and "flat" agrotextile with comprehensive weed control under cover, against the null hypothesis about the lack of such possibility.

## 2. Material and Methods

The field experiment was carried out in 2015–2017 at the Experimental Plant for Cultivar Assessment in Uhnin, Lubelskie Voivodeship (51°34' N, 23°02' E, H = 155 m.a.s.l.), belonging to the Central Research Center for Cultivar Testing. The experiment was carried out by the method of random sub-blocks, in a dependent design (split-split-plot), in three replications. The first order factor were potato cultivars, 'Denar' and 'Lord'. The second order factor were cultivation technologies: (A) traditional technology as a control object, (B) technology with the use of polyethylene film covers, and (C) technology with the use of agrotextile. Factors of the third order were potato weed management practices: (1) mechanical as a control, (2) mechanical and chemical maintenance with the use of Afalon Dispersive 450 SC in the amount of 2.0 $dm^{3.}ha^{-1}$, (3) mechanical–chemical maintenance with the use of the herbicide Racer 250 EC at a dose of 2.0 $dm^{3.}ha^{-1}$, (4) mechanical–chemical maintenance with the use of a mixture of Afalon Dispersive 450 SC and Command 480 EC herbicides in the amount of 1.0 $dm^{3.}ha^{-1}$ and 0.15 $dm^{3.}ha^{-1}$. The size of the plots to be harvested was 15 $m^2$.

### 2.1. Agrotechnical and Plant Protection Treatments

The potato forecrop was winter triticale. After harvesting the forecrop, stubble cultivation was performed. In the fall of each year before planting, winter plowing was carried out to a depth of

about 27 cm. In spring, the field was harrowed, then NPK fertilizers were sown and mixed with the soil with a cultivating unit to a depth of 12 cm. Mineral fertilizers—potassium, phosphorus, and sulfur—were applied to the soil in the following amounts: 39.3 kg·P·ha$^{-1}$, 112.1 kg·K·ha$^{-1}$, and 15.8 kg·S·ha$^{-1}$. The amount of mineral fertilization was determined based on the fertility of the soil with these components. Nitrogen fertilizers were sown in spring in the amount of 90 kg·N·ha$^{-1}$ in the form of polifoska (27 kg·ha$^{-1}$) and urea (63 kg·ha$^{-1}$). Potato propagating material (EU grade A) was planted annually in spring by hand at the end of April, at a spacing of 67.5 cm × 37 cm. The size of a single plot for harvest was 15 m$^2$. After the potato tubers had been planted manually, dredging was carefully performed, combined with light harrowing, to remove the top of the ridges, which is eroded by water and wind. Then, the herbicides were sprayed in the given doses, and the covers were applied. The protection of plants against diseases and pests was carried out in accordance with recommendations of the Institute of Plant Protection, National Research Institute, and the principles of Good Agricultural Practice [9,19]. During the growing season, spraying against alternariosis and potato blight was performed in accordance with the warnings of the Institute of Plant Protection, and the Colorado potato beetle was controlled using available preparations during its occurrence (Table 1).

**Table 1.** The agricultural treatments and plant protection products used in the experiment in the years 2014–2017.

| Autumn 2014–2016 | | |
|---|---|---|
| Tillage | | |
| Winter plowing to a depth of about 27 cm | | |
| Herbicides for forecrop | | |
| Lentipur Flo 500 SC (chlorotoluron) − 1 dm$^3$·ha$^{-1}$ (Autumn 2014) Snajper 600 SC (chlorotoluron + diflufenikan) − 1 dm$^3$·ha$^{-1}$ (Autumn 2014) Glean 75 WG (chlorosulfuron) − 0.01 kg·ha$^{-1}$ (Autumn 2014) Bizon (diflufenikan + florasulam + penoksulam) − 1 dm$^3$·ha$^{-1}$ (Autumn 2015) Lentipur Flo 500 SC (chlorotoluron) − 1 dm$^3$·ha$^{-1}$ (Autumn 2016) Snajper 600 SC (chlorotoluron + diflufenikan) − 1 dm$^3$·ha$^{-1}$ (Autumn 2016) Glean 75 WG (chlorosulfuron) − 0.01 kg·ha$^{-1}$ (Autumn 2016) | | |
| Spring 2015 | Spring 2016 | Spring 2017 |
| Tillage and agricultural treatments | | |
| Harrowing Fertilization of NPK Cultivation with an aggregate Planting of potato seeds—manually Earthing Mechanical weed control Harvest with potato elevator digger | Harrowing Fertilization of NPK Cultivation with an aggregate Planting of potato seeds—manually Earthing Mechanical weed control Harvest with potato elevator digger | Harrowing Fertilization of NPK Cultivation with an aggregate Planting of potato seeds—manually Earthing Mechanical weed control Harvest with potato elevator digger |
| Fungicides—after removing the covers | | |
| Infinito 867.5 SC (propamocarb hydrochloride + fluopicolide) − 1.6 dm$^3$·ha$^{-1}$ Ridomil Gold MZ 67.8 (mancozeb + metalaxyl) − 2 kg·ha$^{-1}$ Infinito 867.5 SC (propamocarb hydrochloride + fluopicolide) − 1.6 dm$^3$·ha$^{-1}$ | Acrobat MZ 69 WG (mancozeb + dimethomorph) − 2 kg·ha$^{-1}$ Infinito 867.5 SC (propamocarb hydrochloride + fluopicolide) − 1.6 dm$^3$·ha$^{-1}$ Acrobat MZ 69 WG mancozeb + dimethomorph) − 2 kg·ha$^{-1}$ | Acrobat MZ 69 WG mancozeb + dimethomorph) − 2 kg·ha$^{-1}$ Infinito 867.5 SC (propamocarb hydrochloride + fluopicolide) − 1.6 dm$^3$·ha$^{-1}$ Acrobat MZ 69 WG mancozeb + dimethomorph) − 2 kg·ha$^{-1}$ |
| Insecticides—after removing the covers | | |
| Apacz 50 WG (clothianidin) − 0.04 kg·ha$^{-1}$ Proteus OD 110 (thiacloprid + deltamethrin) − 0.4 dm$^3$·ha$^{-1}$ | Actara 25 WG (thiamethoxam) − 0.08 kg·ha$^{-1}$ Nuprid 200 SC (imidacloprid) − 0.15 dm$^3$·ha$^{-}$ | Actara 25 WG (thiamethoxam) − 0.08 kg·ha$^{-1}$ Apacz 50 WG (clothianidin) − 0.04 kg·ha$^{-1}$ |

Source: own research.

### 2.2. Herbicide Active Substances

Characteristics of herbicides and adjuvants are given in Table 2.

**Table 2.** Characteristics of the herbicides and herbicide additives used in the experiment.

| Trade Names of the Preparation | Active Substances | Content of Active Substances | Recommended Application Rates per 1 ha | Utility Forms | Grace Period * |
|---|---|---|---|---|---|
| Afalon Dispersive 450 SC | Linuron | $450 \text{ g·dm}^{-3}$ | $1.5–2.0 \text{·dm}^{-3}$ | granules for water suspension | Not applicable |
| Racer 250 EC | Flurochloridone | $250 \text{ g·dm}^{-3}$ | $2.0 \text{·dm}^{-3}$ | concentrate for making a water emulsion | Not applicable |
| Command 480 EC | Clomazone | $480 \text{ g·dm}^{-3}$ | $0.2 \text{·dm}^{-3}$ | concentrate for making a water emulsion | Not applicable |

Sources: Accinelli et al. [12], Caldiz et al. [14], Tomlin [13]; * The period from the last application of the product to the day the crop of potato is harvested.

Afalon Dispersive 450 SC belongs to urea herbicides with toxicity class III. Its biologically active substance is linuron—450 g in 1 kg of the preparation. Linuron [3-(3,4-dichlorocyl)-1-methylurea] is an inhibitor of photosynthesis and electron transport [20,21].

Racer 250 EC is an herbicide of toxicity class IV. The biologically active substance is 250 g of flurochloridone per 1 $dm^3$ of the preparation—[3-chloro-4-chloromethyl-1-(trifluoro-m-tollyl)-2-pyrrolidone. It is an inhibitor of carotenoid synthesis [19,21].

Command 480 EC is a preparation with the toxicity class III. Its biologically active substance is clomazone—480 g in 1 L of the preparation. Clomazone is also known as oxazolide [2-(2-chlorobenzyl)-4,4-dimethyl-oxazolidine-5-one]. It can be mixed with other herbicides, e.g., with Afalon Dispersive (Table 2) [21,22].

*2.3. Soil Sampling and Evaluation*

Before establishing the experiment, 20 primary soil samples were collected each year, constituting one aggregate sample weighing of about 0.5 kg [23]. The samples collected in this way were used to determine the basic physicochemical and chemical properties of the soil. Soil samples were taken from the humus level (0–25 cm), after harvesting the crop, from 20 randomly selected places [24].

Chemical and physicochemical properties of the soil were determined in a certified laboratory of the District Chemical and Agricultural Station in Wesoła near Warsaw by the following methods: soil granulometric composition was determined by laser diffraction Ryzak et al. [25,26]; pH, in a suspension of 1 $\text{mol·KCl·dm}^{-3}$ and in $H_2O$ suspension by potentiometric method [27]; organic carbon content Corg., the Tiurin method [26]; available magnesium content by the Schachtschabel method [28], and content of available forms of phosphorus and potassium (by the Egner-Riehm method) [29,30]. Cu, Mn, Zn, Fe contents in 1 mole HCl were confirmed by the AAS method (Atomic Absorption Spectrometry) [31–33]; B was determined spectrophotometrically with curcumin [34]. The results of soil analysis were valued according to Mocek and Drzymała [24].

*2.4. Soil Analysis*

The experiment was carried out on fallow soil with a sandy loam grain composition WRB [35]. In terms of the percentage of sand, silt, and clay fractions, it is a granulometric subgroup—clay sand (light soil). The sand fraction was 67.0%, the dust fraction was 30.6%, and the clay was 2.4%. This proportion of individual fractions corresponds to the composition of clay dust. In terms of agricultural suitability, these soils belong to the weakly acid good rye complex. This soil is classified in the agronomic category as light mineral [36]. The soil was slightly acidic, low humus content, high to very high phosphorus and magnesium contents, and low to medium potassium content. The soil was characterized by an

average content of manganese and iron, medium to high content of copper, high content of zinc, and high content of boron [24,36] (Table 3).

**Table 3.** Physico-chemical characteristics of the soil before establishing the experiment in 2015–2017.

| Year of Research | Content Macronutrients [mg·100 g⁻¹ soil] | | | Humus Content [g·kg⁻¹] | pH [KCL] | Micronutrients Content [mg·kg⁻¹ soil] | | | | |
|---|---|---|---|---|---|---|---|---|---|---|
| | P | K | Mg | | | Cu | Mn | Zn | Fe | B |
| 2015 | 8.9 | 10.9 | 7.8 | 0.94 | 5.9 | 7.51 | 318 | 40.1 | 3760 | 7.24 |
| 2016 | 8.3 | 9.1 | 7.0 | 1.06 | 5.8 | 4.92 | 337 | 56.7 | 3925 | 5.28 |
| 2017 | 10.6 | 9.8 | 6.3 | 1.03 | 6.6 | 8.99 | 166 | 41.1 | 3600 | 6.04 |
| Mean | 9.3 | 9.9 | 7.0 | 1.02 | | 7.02 | 274 | 46.0 | 3762 | 6.17 |

Source: own results made at the District Chemical and Agricultural Station in Lublin.

*2.5. Assessment of Weed Infestation*

To compare the effectiveness of potato weed management, the infestation was determined by the quadrat. The following were determined: fresh and air-dried mass of weeds, number of monocotyledonous and dicotyledonous weeds, and their floristic composition. Weed infestation analysis was performed at three times: before and after row closure and before potato plants ripened [19].

Before counting the weeds, a list of species was established and entered on the analysis sheet. The square was placed randomly in three places of the plot under evaluation. Weed plants were counted in a box covering an area of 0.5 m². The number of weeds present on an area of 1 m² was calculated according to the formula:

$$Lch = \frac{(L1 + L2 + L3)}{(lp * pr)} \tag{1}$$

where:

$Lch$—number of weeds of a given species growing on an area of 1 m²,
$L_1$; $L_2$; $L_3$—number of plants in the frame in successive measurements (pcs.),
$lp$—number of measurements,
$pr$—frame area (m²) [19].

The fresh weight of all weed species present on the experimental plot was determined. A floristic list of weeds was established, and their names were entered in the analysis sheet. Then samples of the fresh weight of weeds were taken from the area covered by a randomly placed throw square. Plants were taken out of the soil and placed in plastic containers with an inventory label attached. This operation was repeated three times in each plot. The weeds collected in the sample were sorted, grouping individual weed species separately; fresh weight was determined for each species and the weighing results were recorded in the evaluation sheet. The weights of all weed species were summed and the average of three replicates for the plot was calculated. All results were recorded in the worksheet.

The mean fresh weight of weeds was calculated according to the following formula:

$$Sm = \frac{(m1 + m2 + m3)}{3} \tag{2}$$

where:

$Sm$—average weight of the weed species concerned (g),
$m_1$—weight of the weeds in the first throw of the square (g),
$m_2$—weight of weeds in the second square projection (g),
$m_3$—mass of weeds in the third square projection (g).

The determination of the air-dried mass of weeds consisted in manually picking them from the marked places and placing them in a ventilated room until a constant mass was obtained [19].

*2.6. Meteorological Conditions*

Meteorological conditions in the years of the study were varied. The highest amount of rainfall during the potato vegetation period was recorded in 2015, but their distribution was not favorable for the potato because during the intensive plant growth and accumulation of tuber yield, a significant shortage was observed in June–August. The Sielianinov hydrothermal coefficient defines the months of 2016 as: May, quite dry; June, optimal; July, quite humid; August, dry, and September, extremely dry. In the third year of the research, meteorological conditions were changing. Optimal water supply to potato plants was observed alternately, with a huge shortage of them in the subsequent month of vegetation. The Sielianinov hydrothermal coefficient defines the month of May as optimal, June as very dry, July as humid, August as very dry, and September as quite wet (Table 4).

$$k = \frac{10P}{\sum t} \tag{3}$$

**Table 4.** Meteorological conditions in the season of potato vegetation in 2015–2017.

| Year | Month | Rainfall [mm] | | | | Air Temperature [°C] | | | | Hydrothermal Coefficient of Sielianinov * |
|------|-------|---------------|---|---|-------|----------------------|---|---|------|--------------------------|
| | | Decade | | | Month | Decade | | | Mean | |
| | | 1 | 2 | 3 | | 1 | 2 | 3 | | |
| | April | 14.6 | 5.9 | 41.3 | 61.8 | 5.4 | 8.6 | 12.4 | 8.8 | 2.3 |
| | May | 23.4 | 13.9 | 83.0 | 120.3 | 12.6 | 12.0 | 13.7 | 12.8 | 3.0 |
| | June | 5.4 | 16.5 | 24.8 | 46.7 | 17.7 | 16.3 | 16.1 | 16.7 | 0.9 |
| 2015 | July | 10.5 | 21.6 | 13.1 | 45.2 | 19.6 | 18.7 | 19.9 | 19.4 | 0.8 |
| | August | 0.4 | 0 | 5.7 | 6.1 | 23.4 | 20.6 | 20.3 | 21.4 | 0.1 |
| | September | 32.4 | 32.6 | 65.2 | 130.2 | 16.0 | 17.7 | 12.8 | 15.5 | 2.8 |
| | Total | | | | 410.3 | | | | | |
| | April | 11.5 | 22.2 | 13.4 | 47.1 | 10.9 | 10.1 | 9.0 | 10.0 | 1.6 |
| | May | 4.9 | 2.8 | 38.6 | 46.3 | 14.4 | 17.8 | 12.9 | 15.3 | 1.0 |
| | June | 10.1 | 43.2 | 34.0 | 87.3 | 16.6 | 17.5 | 23.0 | 19.1 | 1.5 |
| 2016 | July | 22.4 | 30.8 | 60.9 | 114.1 | 19.5 | 20.1 | 21.9 | 20.5 | 1.8 |
| | August | 22.8 | 17.7 | 0.5 | 41.0 | 20.7 | 17.1 | 20.4 | 19.5 | 0.7 |
| | September | 7.6 | 0.1 | 4.1 | 11.8 | 19.5 | 15.5 | 11.5 | 15.5 | 0.3 |
| | Total | | | | 347.6 | | | | | |
| | April | 6.4 | 7.2 | 38.2 | 51.8 | 10.6 | 6.8 | 6.9 | 8.1 | 2.1 |
| | May | 45.1 | 1.3 | 19.1 | 65.5 | 10.5 | 13.0 | 17.4 | 13.7 | 1.5 |
| | June | 1.9 | 9.2 | 12.0 | 23.1 | 16.6 | 17.7 | 20.7 | 18.3 | 0.4 |
| 2017 | July | 10.1 | 80.9 | 41.0 | 132.0 | 17.9 | 19.0 | 21.0 | 19.4 | 2.2 |
| | August | 0.4 | 24.4 | 2.2 | 27.0 | 22.8 | 21.3 | 17.1 | 20.3 | 0.4 |
| | September | 38.7 | 35.9 | 8.7 | 83.3 | 16.3 | 15.3 | 12.8 | 14.8 | 1.9 |
| | Total | | | | 382.7 | | | | | |

Source: The meteorological station in Uhnin 2015–2017, * coefficient was calculated according to the formula:

Skowera et al. [37].

where P is the sum of the monthly precipitation in mm, Σt is monthly total air temperature > 0 °C. Ranges of values of this index were classified as follows: extremely dry, 0.0 ≤ k < 0.4; very dry, 0.7 ≤ k < 0.4; dry, 1.0 ≤ k < 0.7; rather dry, 1.3 ≤ k < 1.0; optimal, 1.6 ≤ k < 1.3; rather humid, 2.0 ≤ k < 1.6; wet, 2.5 ≤ k < 2.0; very humid, 3.0 ≤ k < 2.5; extremely humid, 3.0 > k.

*2.7. Statistical Calculations*

The obtained results were subjected to statistical analysis. The calculations were performed with the SAS/STAT 9.2 software [38]. They were based on a three-factor analysis of the variance model and multiple t-Tukey tests with the significance level *p* = 0.05. Multiple comparison tests allowed for

detailed analysis of mean comparisons by distinguishing statistically homogeneous groups of mean (homogeneous groups) and determining so-called the smallest significant mean differences, which in Tukey's tests are denoted as HSD (Tukey's honest significant difference) [39].

## 3. Results

### 3.1. Number of Monocotyledonous Weeds

The average population density of monocotyledonous weeds in the potato field was 25.6 pcs. $m^{-2}$. The highest number of weeds of this class was observed before the rows were closed, and the least was before the tuber ripening. On the other hand, their number after row closure and before potato harvest did not differ significantly (Table 5).

**Table 5.** The impact of technology, methods of weed management, cultivars, and years of cultivation on the number of monocotyledonous weeds (pcs $m^{-2}$).

| Factors of the Experiment | | The Terms of Observation * | | | Mean |
|---|---|---|---|---|---|
| | | I | II | III | |
| Technologies | Traditional | 46.8 | 23.0 | 18.2 | 29.3 |
| | PE-Sheeting | 34.7 | 16.4 | 19.9 | 23.7 |
| | PP-Sheeting | 51.0 | 16.9 | 15.9 | 27.9 |
| | HSD$_{0.05}$ | | 10.3 | | ns ** |
| Potato weed management practices | Mechanical | 44.6 | 20.1 | 18.9 | 27.9 |
| | Afalon | 35.5 | 11.4 | 8.5 | 18.5 |
| | Racer | 43.0 | 30.5 | 29.4 | 34.3 |
| | Afalon + Command | 41.2 | 11.9 | 12.7 | 21.9 |
| | HSD$_{0.05}$ | | 12.5 | | 5.7 |
| Cultivars | 'Denar' | 39.9 | 14.4 | 15.9 | 23.4 |
| | 'Lord' | 42.2 | 22.6 | 18.8 | 27.9 |
| | HSD$_{0.05}$ | | ns | | 3.1 |
| Years | 2015 | 23.3 | 33.5 | 42.7 | 33.2 |
| | 2016 | 18.6 | 18.1 | 5.5 | 14.1 |
| | 2017 | 80.8 | 3.9 | 4.5 | 29.7 |
| | HSD$_{0.05}$ | | 10.3 | | 5.1 |
| Mean | | 41.1 | 18.5 | 17.4 | 25.6 |
| HSD$_{0.05}$ | | | 5.1 | | |

* I, before shorting rows; II, after closing the rows; III, before potato ripening; ** differences not significant at p$_{0.05}$ level.

Only the cultivation technology under a polyethylene film cover, compared to traditional cultivation, significantly reduced the number of weeds in this group, but only before the potato row closing. Weed control methods with Afalon Dispersive 450 SC and a mixture of herbicides Afalon Dispersive 450 SC and Command 480 EC significantly reduced the number of monocotyledonous weeds compared to mechanical control. The greatest decrease in the number of weeds was observed in mechanical and chemical methods using the herbicide Afalon Dispersive 450 S.C. Only potato methods with Racer 250 EC contributed to an increase in the number of monocotyledonous weeds compared to mechanical method. Such significant increase was observed only before the potato harvest. The potato weed control methods did not differentiate the population density of monocotyledonous weeds before the rows were closed, while after the rows closing and before the tuber ripening, development of this group of weeds was most limited by the treatment with Afalon Dispersive 450 S.C. The objects where 'Denar' cv. was grown were characterized by a smaller population density of monocotyledonous weeds than those with 'Lord' cv., which results from greater foliage and better shading of the soil by plants of the former cultivar. Regardless of the experimental factors, the randomness factor played an

important role in shaping the population density of monocotyledonous weeds. The lowest number of weeds in this group was recorded in warm 2016, with optimally distributed rainfall during the growing season. Most of them were recorded in 2017, with periodic excesses and deficiencies of rainfall during the growing season and with air temperatures higher than the long-term average. The number of monocotyledonous weeds also depended on the time of observation. Their highest population density was found in 2017 before the potato rows were closed, and then it was drastically reduced (more than 20 times). In 2016, the reduction in weeds in this group took place only before the tuber ripening. On the other hand, in 2015, when there was a significant shortage of rainfall and high air temperatures during the period of intensive growth of potato plants and accumulation of tuber yield, an increase in the number of monocotyledonous weeds was observed as the plants developed, and their greatest number was found before the potato ripening (the so-called secondary weed infestation) (Table 5).

### 3.2. Number of Dicotyledonous Weeds

Population density of dicotyledonous weeds was on average 20.2 $PLA/m^{-2}$. The greatest weed infestation of the potato plantation with this group of weeds occurred before the rows were closed, while the smallest after their closing (Table 6).

**Table 6.** The impact of technology, methods of weed management, cultivars, and years on the number of dicotyledonous weeds ($PLA/m^{-2}$).

| Factors of the Experiment | | Terms of Observation * | | | Mean |
|---|---|---|---|---|---|
| | | I | II | III | |
| Technologies | Traditional | 27.3 | 15.1 | 16.4 | 19.6 |
| | PE-Sheeting | 18.6 | 18.5 | 24.6 | 20.6 |
| | PP-Sheeting | 24.4 | 17.9 | 20.0 | 20.8 |
| | $HSD_{0.05}$ | | ns ** | | ns |
| Potato weed management practices | Mechanical | 26.3 | 11.5 | 17.5 | 18.4 |
| | Afalon | 32.1 | 21.5 | 19.0 | 24.2 |
| | Racer | 25.4 | 20.3 | 25.7 | 23.8 |
| | Afalon + Command | 17.7 | 11.5 | 14.4 | 14.5 |
| | $HSD_{0.05}$ | | ns | | 5.3 |
| Cultivars | 'Denar' | 27.4 | 15.6 | 18.9 | 20.6 |
| | 'Lord' | 23.4 | 16.7 | 19.3 | 19.8 |
| | $HSD_{0.05}$ | | ns | | ns |
| Years | 2015 | 9.3 | 26.3 | 38.8 | 24.8 |
| | 2016 | 22.2 | 14.3 | 10.3 | 15.6 |
| | 2017 | 44.1 | 8.3 | 8.7 | 20.4 |
| | $HSD_{0.05}$ | | 9.6 | | 4.2 |
| Mean | | 25.4 | 16.2 | 19.1 | 20.2 |
| $HSD_{0.05}$ | | | 4.2 | | |

* I, before shorting rows; II, after closing the rows; III, before potato ripening; ** differences not significant at $p_{0.05}$ level.

Cultivation technologies did not differentiate the value of this feature in any of the weed infestation observation periods. Reduction in the number of dicotyledonous weeds was facilitated using a mixture of Afalon Dispersive 450 SC and Command 480 EC and mechanical method. Destruction of weeds with the use of Afalon Dispersive 450 SC and Racer 250 EC turned out to be homogeneous in this respect. Variable meteorological conditions during the potato growing season, regardless of the experimental factors, differentiated the value of this feature. The lowest number of dicotyledonous weeds was observed in 2016 with an optimally distributed amount of rainfall over time, and the highest in 2015 with significant shortage of rainfall and high air temperatures during the potato growing season. The number of weeds in this group depended on the interaction of the development phases of potato plants

and meteorological conditions in the years of the study. In 2015, with a late and cool spring, the number of dicotyledonous weeds increased until the end of potato vegetation. In 2016, the highest number was recorded before the potato row closing, and then it was gradually reduced. In 2017, in a warm spring, the highest number of weeds was recorded before the rows were closed and after their closing; it was reduced by as much as 5 times and only slightly increased before tuber ripening (Table 6).

### 3.3. Fresh Mass of Weeds

The lowest weed mass was recorded before the potato rows closing, and the highest one before the plants ripened, which was caused by secondary weed infestation (Table 7).

**Table 7.** Impact of technology, methods of weed management, cultivars, and years of cultivation on fresh weed mass (g m$^{-2}$).

| Factors of the Experiment | | Terms of Observation * | | | Mean |
|---|---|---|---|---|---|
| | | I | II | III | |
| Technologies | Traditional | 37 | 63 | 203 | 101 |
| | PE-Sheeting | 24 | 66 | 211 | 100 |
| | PP-Sheeting | 29 | 48 | 203 | 93 |
| | HSD$_{0.05}$ | | 28 | | ns ** |
| Potato weed management practices | Mechanical | 21 | 55 | 201 | 92 |
| | Afalon | 27 | 53 | 170 | 83 |
| | Racer | 36 | 43 | 270 | 116 |
| | Afalon + Command | 21 | 72 | 185 | 93 |
| | HSD$_{0.05}$ | | 34 | | 15 |
| Cultivars | 'Denar' | 26 | 50 | 220 | 99 |
| | 'Lord' | 27 | 62 | 193 | 94 |
| | HSD$_{0.05}$ | | ns | | ns |
| Years | 2015 | 13 | 58 | 482 | 184 |
| | 2016 | 5 | 15 | 105 | 42 |
| | 2017 | 61 | 95 | 36 | 64 |
| | HSD$_{0.05}$ | | 28 | | 12 |
| Mean | | 27 | 56 | 207 | 96 |
| HSD$_{0.05}$ | | | 12 | | |

* I, before shorting rows; II, after closing the rows; III, before potato ripening; ** differences not significant at p$_{0.05}$ level.

The use of potato cultivation technology with coverings did not significantly affect the value of this feature. Only in cultivation under a polyethylene film cover, a successive increase in the fresh weight of weeds was observed as the potato plants developed. In traditional cultivation, as well as in objects with the use of polypropylene agrotextile as a cover, a significant difference in the value of this feature occurred between the first and the last observation of weed infestation. The use of Afalon Dispersive 450 SC and a mixture of Afalon Dispersive 450 SC and Command 480 EC did not significantly reduce the fresh weight of weeds per unit area, compared to the traditional, mechanical method of controlling the segetal vegetation. Moreover, both these methods of method turned out to be homogeneous in terms of the value of this feature. On the other hand, potato method with the Racer 250 EC herbicide appeared to be ineffective and even contributed to an increase in the fresh mass of weeds, compared to the mechanical method, considered a standard object, and to other weed management methods. Morphological features of studied cultivars did not differentiate the fresh mass of weeds in the field, while meteorological conditions in the years of study significantly modified this feature. The lowest fresh mass of weeds was recorded in 2016 with warm and optimally distributed rainfall during the growing season, and the highest was in 2015 with a significant shortage of rainfall and air temperature higher than the multi-annual average (Table 7).

### 3.4. Air-Dried Mass of Weeds

Phases of plant development were the factor determining the amount of air-dried mass of weeds in a potato field. The air-dried mass of weeds was the lowest before the rows were closed, and the highest was before potato harvesting (Table 8).

**Table 8.** Impact of cultivation technology, methods of weed management, cultivars, and years of cultivation on the air-dried mass of weeds (g. $m^{-2}$).

| Factors of the Experiment | | Observation Dates * | | | Mean |
|---|---|---|---|---|---|
| | | I | II | III | |
| Technologies | Traditional | 8 | 11 | 82 | 34 |
| | PE-Sheeting | 5 | 13 | 89 | 36 |
| | PP-Sheeting | 6 | 9 | 107 | 41 |
| | HSD$_{0.05}$ | | 19 | | ns ** |
| Potato weed management practices | Mechanical | 4 | 9 | 89 | 34 |
| | Afalon | 5 | 9 | 82 | 32 |
| | Racer | 7 | 9 | 112 | 43 |
| | Afalon + Command | 4 | 15 | 83 | 34 |
| | HSD$_{0.05}$ | | 19 | | 9 |
| Cultivars | 'Denar' | 5 | 10 | 92 | 36 |
| | 'Lord' | 5 | 11 | 91 | 36 |
| | HSD$_{0.05}$ | | ns | | ns |
| Years | 2015 | 2 | 11 | 101 | 38 |
| | 2016 | 1 | 2 | 31 | 11 |
| | 2017 | 12 | 18 | 143 | 58 |
| | HSD$_{0.05}$ | | ns | | 7 |
| Mean | | 6 | 10 | 92 | 36 |
| HSD$_{0.05}$ | | | 7 | | |

* I, before shorting rows; II, after closing the rows; III, before potato ripening; ** differences not significant at p$_{0.05}$ level.

Technologies of potato cultivation and varietal properties did not significantly differentiate this feature. However, a significant interaction of potato cultivation technology and dates of weed infestation observation was found. In the cultivation under cover of polypropylene agrotextile, a significant increase in the air-dried mass of weeds was observed, compared to the traditional technology, but only before the potato was mature. Among the methods of mechanical and chemical weed control used in the experiment, the use of the Racer 250 EC preparation contributed to a significant increase in the dry mass of weeds, compared to both the standard object and other methods of potato cultivation. The latter turned out to be homogeneous in terms of the value of this feature. However, this negative effect of method with Racer 250 EC was noticed only in the period before tuber ripening, i.e., regarding secondary weed infestation. Meteorological conditions during the growing season had a decisive influence on the value of this feature. The lowest air-dried mass of weeds was recorded in 2016, which was the most optimal, both in terms of rainfall and air temperature, and the highest value of this characteristic was recorded in 2017, when there were periodical excesses or deficiencies of rainfall during the potato growing season (Table 8).

### 3.5. Weed Species Composition

In the potato field, seven species of monocotyledonous weeds and 20 species of dicotyledonous weeds were recorded. Among monocotyledonous weeds, the most numerous species were: *Echinochloa crus-galli*, *Setaria glauca*, and *Setaria viridis*, and only occasionally, *Avena fatua* and *Poa annua*. The highest number of *Echinochloa crus-galli* was recorded before the row closing, and the lowest one was before the potato was ripened. In turn, the highest numbers of *Setaria viridis* and *Setaria glauca*

were observed before the potato ripening, which was related to the development biology of these species. Cultivation technologies used significantly modified the number of *Echinochloa crus-galli* and *Setaria glauca* species. The agrotextile cover, for *Echinochloa crus-galli*, contributed to the increase in the number of this species, compared to the foil cover, but did not differ significantly from the abundance of this species in traditional technology. It follows that not only crops, but also weeds, and especially thermophilic species, have better growth and development conditions under the agrotextile cover. The number of *Echinochloa crus-galli*, the most onerous species of the monocotyledonous species, was most effectively limited by mechanical and chemical treatment with Afalon Dispersive 450 SC, compared to mechanical treatment. The use of the Afalon Dispersive 450 SC and Command 480 EC herbicide mixture significantly reduced the number of *Echinochloa crus-galli* per unit area, while the use of the Racer 250 EC herbicide did not reduce the weed infestation with this species, and there was even a tendency to increase its number under the influence of this herbicide compared to the standard facility. In the objects with 'Denar' cv., the species *Echinochloa crus-galli* in the field was less numerous than in the combinations with the 'Lord' cv. It should be assumed that this was related to the faster growth rate of 'Denar' cv. and more leafy conformation of these plants, which probably contributed to faster and better soil protection by the cultivated plant. The most significant variation in the number and species composition of weeds occurred in the years of the study. This was due to the meteorological conditions of the growing season, differences in the years of the study, different abundances of available nutrients in the soil—especially potassium, phosphorus, calcium, and magnesium—and the weed seed abundance that varied between the years. In 2016, there were only three species of monocotyledonous weeds, while in the warm and alternately severe and dry 2017, there were as many as seven species, including *Echinochloa crus-galli*, in large numbers. While an abundant species composition of monocotyledonous weeds was observed in the primary weed infestation, only *Echinochloa crus-galli*, *Setaria glauca*, and *Setaria viridis* occurred in the secondary weed infestation. The most abundant species of dicotyledonous weeds turned out to be *Anthemis arvensis* and *Polygonum convolvulus*, and less abundantly in the soil were *Viola arvensis*, *Veronica hederaefolia*, *Cirsium arvense*, and *Stellaria media*, while the remaining species occurred sporadically. Mechanical weed control only limited the incidence of *Anthemis arvensis*. The application of mechanical and chemical treatment with the herbicide Afalon Dispersive 450 SC reduced the number of *Stellaria media*, *Galium aparine*, *Centaurea cyanus*, and *Anagalis arvensis*, and it was ineffective against *Anthemis arvensis*, *Polygonum convolvulus*, and *Veronica hedereafolia*. In turn, weed management methods with Racer 250 EC eliminated *Polygonum lapathifolium* and *Galium aparine*, and it limited *Polygonum convolvulus* and *Viola arvensis*. Mechanical and chemical treatment with the use of a mixture of Afalon Dispersive 450 SC and Command 480 EC preparations limited the occurrence of the following species: *Chenopodium album*, *Cirsium arvense*, *Spergula arvensis*, and *Vicia tetrasperma*. Variable meteorological and soil conditions in the years of research also modified the number of dicotyledonous weeds. In 2015—which was poor in rainfall, but warm and sunny—the largest number of the following species was recorded: *Anthemis arvensis*, *Stellaria media*, *Veronica hedereafolia*, *Vicia tetrasperma*, and *Viola arvensis*. In 2016, the most optimal in terms of temperature and rainfall were the most numerous species requiring thermal and wetness conditions—*Cirsium arvense*, *Polygonum lapathifolium*, *Polygonum aviculare*, *Spergula arvensis*, *Raphanus raphanistrum*, and *Rumex acetosella*—while in 2017, when a periodic excess or a shortage of precipitation was observed, the most numerous were *Chenopodium album*, *Polygonum convolvulus*, *Galeopsis tetrahit*, and *Vicia hirsute*. The species composition and the number of dicotyledonous weeds also varied over time. Before the rows of the crop plant were closed, the most numerous were *Chenopodium album*, *Raphanus raphanistrum*, and *Polygonum convolvulus*. After row closure, *Cirsium arvense* was the most abundant species, and before the potato ripening, it was *Anthemis arvensis*, *Spergula arvensis*, *Stellaria media*, *Viola arvensis*, *Veronica hedereafolia* and *Myosotis arvensis*, which results from the phenology of weeds (Table 9). It should be noted that there were no weeds from other botanical groups in the species composition.

**Table 9.** Species composition and number of mono- and dicotyledonous weeds (pcs m$^{-2}$).

| Factors of the Experiment | | *Agropyron repens* | *Apera spica-venti* | *Avena fatua* | *Echinochloa crus-galli* | *Poa annua* | *Setaria glauca* | *Setaria viridis* | *Anagalis arvensis* | *Anthemis arvensis* | *Centaurea cyanus* | *Chenopodium album* | *Cirsium arvense* | *Galium aparine* | *Galeopsis tetrahit* | *Myosotis arvensis* | *Poligonum convolvulus* | *Poligonum lapathifolium* | *Poligonum aviculare* | *Stellaria media* | *Spergula arvensis* | *Veronica hederaefolia* | *Vicia hirsuta* | *Vicia cracca* | *Vicia tetrasperma* | *Viola arvensis* | *Raphanus raphanistrum* | *Rumex acetosella* |
|---|---|---|---|---|---|---|---|---|---|---|---|---|---|---|---|---|---|---|---|---|---|---|---|---|---|---|---|---|
| Technologies | Traditional | 0.6 | 0.3 | 0.2 | 24.4 | 0.2 | 2.6 | 1.1 | 0.1 | 5.1 | 0.9 | 4.5 | 0.5 | 0.0 | 0.0 | 0.4 | 1.5 | 0.4 | 0.1 | 0.5 | 0.7 | 1.0 | 0.1 | 0.1 | 0.2 | 1.3 | 2.9 | 0.3 |
| | PE-Sheeting | 0.6 | 0.2 | 0.1 | 19.8 | 0.2 | 1.8 | 1.1 | 0.1 | 5.9 | 0.9 | 4.5 | 1.2 | 0.0 | 0.0 | 0.3 | 1.7 | 0.3 | 0.3 | 0.6 | 0.8 | 0.7 | 0.1 | 0.0 | 0.3 | 0.9 | 3.2 | 0.1 |
| | PP-Sheeting | 0.8 | 0.1 | 0.0 | 26.3 | 0.0 | 0.2 | 1.2 | 0.2 | 5.8 | 0.6 | 4.4 | 0.7 | 0.1 | 0.1 | 0.2 | 1.7 | 0.1 | 0.2 | 0.5 | 0.4 | 0.6 | 0.1 | 0.0 | 0.3 | 0.5 | 5.5 | 0.1 |
| | HSD$_{0.05}$ | ns * | ns | ns | 5.6 | ns | 0.4 | ns | 0.1 | ns | ns | ns | 0.6 | ns | ns | 0.2 | ns | 0.2 | 0.2 | ns | 0.7 | 0.5 | ns | ns | ns | 0.5 | 2.1 | ns |
| Potato weed management practices | Mechanical | 0.7 | 0.1 | 0.1 | 24.4 | 0.2 | 1.5 | 1.4 | 0.1 | 3.0 | 1.0 | 5.3 | 0.6 | 0.0 | 0.2 | 0.2 | 1.9 | 0.1 | 0.1 | 0.6 | 0.4 | 0.4 | 0.1 | 0.0 | 0.3 | 0.6 | 4.4 | 0.1 |
| | Afalon | 0.5 | 0.1 | 0.0 | 16.5 | 0.0 | 0.6 | 0.7 | 0.1 | 8.8 | 0.6 | 4.0 | 0.6 | 0.0 | 0.1 | 0.4 | 2.4 | 0.3 | 0.2 | 0.4 | 0.6 | 1.4 | 0.1 | 0.1 | 0.6 | 1.1 | 2.9 | 0.3 |
| | Racer | 0.5 | 0.3 | 0.1 | 29.8 | 0.1 | 1.9 | 1.6 | 0.2 | 6.7 | 0.8 | 9.0 | 1.2 | 0.0 | 0.0 | 0.4 | 1.0 | 0.0 | 0.1 | 0.6 | 0.9 | 0.6 | 0.0 | 0.0 | 0.3 | 0.6 | 2.9 | 0.2 |
| | Afalon + Command | 0.5 | 0.2 | 0.1 | 18.9 | 0.0 | 0.8 | 1.0 | 0.1 | 3.9 | 0.9 | 1.9 | 0.5 | 0.1 | 0.0 | 0.2 | 1.9 | 0.4 | 0.2 | 0.9 | 0.1 | 0.3 | 0.0 | 0.1 | 0.1 | 1.3 | 2.6 | 0.1 |
| | HSD$_{0.05}$ | 0.5 | ns | ns | 5.6 | ns | 0.4 | 0.7 | 0.1 | 2.3 | ns | 3.1 | 0.6 | 0.1 | 0.1 | 0.2 | 1.1 | 0.2 | n | 0.4 | 0.7 | 0.5 | ns | ns | 0.3 | 0.5 | ns | ns |
| Cultivars | 'Denar' | 0.7 | 0.2 | 0.1 | 20.6 | 0.1 | 1.1 | 1.0 | 0.1 | 5.4 | 0.7 | 4.7 | 0.7 | 0.0 | 0.0 | 0.4 | 1.9 | 0.3 | 0.1 | 0.7 | 0.4 | 0.5 | 0.1 | 0.1 | 0.4 | 0.8 | 3.2 | 0.1 |
| | 'Lord' | 0.8 | 0.1 | 0.1 | 24.2 | 0.1 | 1.3 | 1.3 | 0.1 | 5.7 | 0.9 | 5.4 | 0.7 | 0.0 | 0.1 | 0.2 | 1.7 | 0.1 | 0.2 | 0.6 | 0.6 | 0.9 | 0.1 | 0.0 | 0.3 | 1.0 | 3.2 | 0.2 |
| | HSD$_{0.05}$ | ns | ns | ns | 3.0 | ns | ns | ns | ns | ns | ns | 0.2 | ns | ns | ns | 0.1 | 0.1 | ns | 0.1 | ns | ns | 0.3 | ns | ns | ns | ns | ns | ns |
| Years | 2015 | 0.3 | 0.5 | 0.3 | 25.8 | 0.3 | 3.1 | 2.9 | 0.2 | 13.5 | 1.2 | 3.4 | 0.1 | 0.0 | 0.0 | 0.6 | 1.1 | 0.0 | 0.0 | 1.2 | 0.3 | 2.0 | 0.0 | 0.0 | 1.0 | 1.3 | 0.0 | 0.0 |
| | 2016 | 1.6 | 0.0 | 0.0 | 12.5 | 0.0 | 0.6 | 0.0 | 0.0 | 0.0 | 0.0 | 2.7 | 2.1 | 0.1 | 0.0 | 0.1 | 0.2 | 0.5 | 0.5 | 0.0 | 1.1 | 0.0 | 0.0 | 0.1 | 0.0 | 0.2 | 8.7 | 0.5 |
| | 2017 | 0.3 | 0.0 | 0.0 | 29.0 | 0.0 | 0.0 | 0.5 | 0.1 | 0.0 | 1.0 | 9.1 | 0.0 | 0.0 | 0.2 | 0.2 | 4.1 | 0.1 | 0.0 | 0.6 | 0.1 | 0.0 | 0.2 | 0.0 | 0.0 | 1.2 | 0.9 | 0.0 |
| | HSD$_{0.05}$ | 0.4 | 0.2 | 0.2 | 4.4 | 0.2 | 0.3 | 0.5 | 0.1 | 1.8 | 0.4 | 2.5 | 0.4 | 0.1 | 0.1 | 0.2 | 0.9 | 0.2 | 0.1 | 0.3 | 0.5 | 0.4 | 0.1 | 0.1 | 0.2 | 0.4 | 1.7 | 0.3 |
| Term of Observations * | I | 0.9 | 0.1 | 0.0 | 39.9 | 0.0 | 0.0 | 0.0 | 0.1 | 4.1 | 0.7 | 7.8 | 0.1 | 0.0 | 0.1 | 0.1 | 3.2 | 0.2 | 0.1 | 0.4 | 0.1 | 0.3 | 0.2 | 0.0 | 0.4 | 1.1 | 6.1 | 0.2 |
| | II | 0.6 | 0.1 | 0.1 | 17.1 | 0.3 | 0.0 | 0.8 | 0.0 | 6.3 | 0.8 | 3.8 | 1.3 | 0.0 | 0.0 | 0.0 | 0.5 | 0.1 | 0.3 | 0.2 | 0.1 | 0.8 | 0.0 | 0.1 | 0.2 | 0.4 | 2.9 | 0.0 |
| | III | 0.7 | 0.3 | 0.2 | 10.2 | 0.1 | 3.7 | 2.7 | 0.2 | 6.4 | 0.9 | 3.5 | 0.8 | 0.1 | 0.0 | 0.7 | 1.7 | 0.2 | 0.2 | 1.3 | 1.4 | 0.9 | 0.0 | 0.0 | 0.4 | 1.2 | 0.6 | 0.3 |
| | HSD$_{0.05}$ | ns | 0.2 | 0.2 | 4.4 | 0.2 | 0.3 | 0.5 | 0.1 | 1.8 | ns | 2.5 | 0.4 | 0.1 | ns | 0.2 | 0.9 | ns | 0.1 | 0.3 | 0.5 | 0.4 | 0.1 | 0.1 | ns | 0.4 | 1.7 | ns |
| Mean | | 0.7 | 0.2 | 0.1 | 22.4 | 0.1 | 1.2 | 1.2 | 0.1 | 5.6 | 0.8 | 5.0 | 0.7 | 0.0 | 0.0 | 0.3 | 1.8 | 0.2 | 0.2 | 0.6 | 0.5 | 0.7 | 0.1 | 0.0 | 0.3 | 0.9 | 3.2 | 0.2 |

* insignificant differences at p$_{0.05}$ level, before row closing; after shorting the rows; before ripening potato.

## 4. Discussion

The conducted research proved that the degree of soil coverage with weeds, both monocotyledonous and dicotyledonous plants in covered objects, was not significantly higher than in traditional cultivation without cover. These results are confirmed by Pszczółkowski and Sawicka [3]. Lutomirska and Roztropowicz [40], Lutomirska [4], Wadas [2], Krzysztofik [1], and Sawicka [41] are of different opinion. These authors prove that the better development of weeds under cover results from the fact that the temperature of the soil under cover, in comparison with the open area, is 1–2 °C higher; moreover, air saturation with water vapor is much higher than in the open area. This difference can be explained by the fact that conditions under the covers favor the germination and growth of weeds, and high saturation with water vapor under cover promotes the formation of the so-called herbicide filter on the surface of ridge, which eliminates germinating zoospores, and the fact that strict field research was carried out in the years with air and soil temperatures significantly exceeding the long-term average for the research point.

Own research confirmed previous reports by Gugała et al. [7], Azadbakht et al. [10], and Baranowska et al. [11] that the number of weeds increases until the potato is fully flowering, and in the later stages it decreases, while their air-dried mass increases until the end of the potato vegetation. Technologies of cultivation with the use of covers did not significantly differentiate this feature, but it was modified by the methods of weed management. The highest fresh and dry mass of weeds was observed during the full ripeness period of the potato, which was caused by secondary weed infestation, which is determined, as reported by Zarzecka et al. [12], by the length of the period of vegetation and the aboveground potato mass. Sawicka and Pszczółkowski [42] proved that the highest tolerated weed infestation, expressed in dry mass of weeds, in the cultivation of very early potato cultivars on light soils at the end of potato vegetation, for the total yield it amounts to 98 g, in mechanical and chemical treatment with the use of herbicide Afalon 50 WP it was 64 g, in the method with Racer 25 EC it was 135 g, and in the method with the Afalon 50 WP and Command 480 EC herbicide mixture it was 204 g·m$^{-2}$. Cultivation with the mixture of herbicides Afalon Dispersive 450 SC and Command 480 EC gave a similar effect in destroying the fresh weed mass as their mechanical control. The research by Gugała et al. [7] shows that the mechanical and chemical method reduces the weight of weeds by about 59%. Ciesielska and Wysmułek [43] confirm this view as well.

In cultivation under cover, mechanical method treatments can be performed only after planting, before placing the covers, thus supporting this method of weed control with chemical treatments is the only way to reduce the weed infestation in the potato field [1,42]. The use of herbicides in potato cultivation immediately after planting in combination with previous mechanical method did not result in complete elimination of weeds. Weed species that survive after the treatment may constitute, in the opinion of Sawicka et al. [44], Barbaś [5], Gugała et al. [7], and Zarzecka [12], a great threat to the cultivated plant, as it may contribute to significant losses of tuber yield. Hess and Foy [18] report the interaction of surfactants with plant skin and possible damage to it.

The total number of weeds per unit area in the field of cultivated potato cultivars exceeded the weed harmfulness threshold adopted by Pszczółkowski and Sawicka [45]. Their number was most severely limited employing a mixture of Afalon Dispersive 450 SC and Command 480 EC.

According to Accinelli et al. [20], Grundy et al. [46], and Haliniarz [47], the complete elimination of weeds from crops is impossible, as they are an inseparable component of agrocenoses. In the opinion of Zarzecka and Gugała [48], each crop may contain a certain number of weeds, which do not reduce its yield. According to Zarzecka et al. [12], the complete elimination of weeds from crops is not possible, because they are an inseparable component of agrocenoses, but their occurrence can be limited to a level that does not exceed the harmfulness threshold, without reducing the yield. In the opinion of Gugała et al. [7], the limit of harmless weed infestation should be defined as "tolerated weed infestation" and, according to Barralis [49], "critical density".

Results of the author's own research stating that the weather conditions, varied in particular years of research, significantly affect the population density of weeds, and the amount of fresh and

air-dried weed mass are consistent with the reports of many authors [2,7,12,42,50], as in recent years, abiotic stresses have had a large impact on the growth and productivity of crops. Sawicka et al. [51] and Wadas and Dziugieł [52] proved that in regions with extensive plant production, there are more and more frequent periods of high temperature and drought, which disrupt photosynthesis and limits the potential of potato yielding. Krzyżewska et al. [53] proved that in Poland, the most unfavorable bioclimatic conditions occur in the south-east of Poland, where forceful heat stress occurs with a frequency of 10%. This phenomenon is associated with a drastic reduction in yields, especially of root crops, as well as with the development of drought-resistant segetal species. The highest number of cases with strong and very strong heat stress was recorded in 2015, and heat waves were observed in the first half of August. This is an important period for the accumulation of yield from root crops. The authors recorded an increase in the number of days with strong and forceful heat stress and their highest frequency in July. Inhibition of potato plant growth is observed at the temperature of 29 °C [41], but most dicotyledonous and monocotyledonous species are resistant to high air temperatures and survive this period, and after a drought period, they increase their weight and density, taking advantage of the inhibition of crop growth. Simultaneously, the increased number of weeds uses the same water resources as the cultivated plants and limits the access of light to them [47].

Diversity of the number and species composition of weeds was also determined by the course of weather conditions in the years of the study. Their higher numbers and more abundant species composition were observed in warm and dry year than in cooler and wetter years. These results are consistent with the observations of Gugała et al. [7], Baranowska et al. [11], and Barbaś and Sawicka [54].

Our research proved that the morphological properties of tested cultivars and length of their growing season determined only the number of monocotyledonous weeds, while Gugała et al. [55,56] found that the physiological and morphological properties of cultivars, i.e., conformation of bushes, their foliage, and position of leaves relating to the light, are factors that most strongly modify the weed mass.

The species composition of weeds in the field of the studied potato cultivars was quite abundant, even though the experiment was conducted on light soil (seven monocotyledon species and 20 dicotyledonous species). Stešević and Jovović [57], in the Lublin region Haliniarz [47], made similar observations. They observed 36–48 species of weeds in potato cultivation, but only five to eight species were dominant. The most abundant of the monocotyledons was *Echinochloa crus-galli*, and of the dicotyledons, it was *Chenopodium album*, *Anthemis arvensis*, and *Raphanus raphanistrum*. Haliniarz [47] came to similar conclusions.

The species composition of weeds was significantly modified by the applied treatments. Afalon Dispersive 450 SC used in the mechanical and chemical control of weeds showed high efficiency in reducing *Echinochloa crus-galli*, *Setaria glauca*, *Setaria viridis*, *Centaurea cyanus*, *Stellaria media*, *Anagalis arvensis*, and no or weak action against *Anthemis arvensis*, *Polygonum convolvulus*, *Vicia tetrasperma*, and *Veronica hederaefolia*. Accinelli et al. [20], Andreasen and Streibig [58], and Gugała et al. [55] recorded similar efficiency of Afalon 50 WP herbicide in other formulation. According to the research of Haliniarz [47], the following weed species are particularly high in nutrient uptake from the soil: *Stellaria media*, *Amaranthus retroflexus*, *Echinochloa crus-galli*, and *Chenopodium album*. Considering their common occurrence, these species should be considered particularly competitive for potato cultivars grown under cover; moreover, the uptake of minerals from the soil by weeds exceeds the uptake by cultivated plants several times and reaches over 80% of macroelements and over 90% of their total uptake by weeds and the plant being grown.

Herbicide Racer 250 EC used in the care of potatoes turned out to be effective against *Polygonum convolvulus*, *Polygonum lapathifolium*, *Galium aparine*, and *Viola arvensis* and ineffective against *Echinochloa crus-galli*, *Setaria glauca*, *Setaria viridis*, *Chemopodium album*, *Cirsium arvensis*, *Spergula arvensis*, and *Mysostis arvensis*. However, it did not limit the fresh and dry mass of weeds, which mainly applies to

secondary weed infestation. Similar results were obtained by Sawicka and Pszczółkowski [42], which is consistent with the reports of Gupta and Gajbhiye [59].

Our own research confirms the reports of Barbaś [5] and Zarzecka et al. [48,60], with high efficiency of herbicide mixtures used in potato cultivation. The use of a mixture of Afalon Dispersive 450 SC and Command 480 EC in potato cultivation showed high efficiency in reducing such species as *Chenopodium album*, *Raphanus raphanistrum*, *Vicia tetrasperma*, *Cirsium arvense*, *Galeopsis tetrahit*, *Spergula arvensis*, *Veronica hederaefolia*, and *Myosotis arvensis*, while it showed weak action towards *Polygonum lapathifolium*, *Galium aparine*, *Viola arvensis*, and *Stellaria media*. Mystkowska et al. [61] and Mayerová et al. [62] showed that both the application of mechanical and chemical treatments, with the use of single herbicides and their mixtures, significantly reduced the weed infestation in potato crops. Research by Riethmuller-Haage et al. [63] demonstrated the importance of the sensitivity of herbicide combinations and species and proved that pre-spray weather information is crucial in developing the dose reduction recommendations.

In the conducted studies, the phytotoxic effect of the herbicides used was observed not only on the weeds to be controlled, but also on the cultivated plant. To provide potato plants with the best possible conditions for growth and development, according to Gugała et al. [7], along with the application of herbicides, biostimulators should also be used, which, in their opinion, reduces the phytotoxic effect of herbicides on potato plants and significantly reduces the fresh mass of weeds, compared to the standard object. According to Eshel et al. [64], cultivation under cover (CC) has direct benefits to a society by reducing the cost of agricultural damage and restoring the infrastructure, and by protecting the environmental benefits provided by agriculture.

According to Wang and Liu [65], Haliniarz [47], Barbaś and Sawicka [54], and Zarzecka et al. [12], the occurrence or significant delay in the resistance of weeds to herbicides is prevented, among others, by crop rotation; alternating the use of agents from various chemical groups, as well as the use of mixtures of herbicides with different mechanisms of action; the application of herbicides to weeds in the period of their greatest sensitivity; the application of herbicides in doses guaranteeing complete destruction of weeds; the addition of adjuvants to the spray liquid in case of dose reduction, including mechanical treatments in the weed control system, and the use of non-selective herbicides before the emergence of the crop.

Chemical methods of weeding with the use of the tested preparations reduced the population of weeds to a varying degree, depending on the herbicide used. Afalon Dispersive 450 SC showed high efficiency in reducing *Echinochloa crus-galli*, *Setaria glauca*, *Setaria viridis*, *Centaurea cyanus*, *Stellaria media*, and *Anagalis arvensis* and showed no or weak activity against *Anthemis arvensis*, *Polygonum convolvulus*, *Vicia tetraspderaolia*, and *Vicia tetraspderaolia*. Racer 250 EC herbicide proved to be effective in reducing *Polygonum convolvulus*, *Polygonum lapathifolium*, *Galium aparine*, and *Viola arvensis* and was ineffective against *Echinochloa crus-galli*, *Setaria glauca*, *Setaria viridis*, *Chemopodium album*, *Cirsium arvensis*, *Spergula arvensis*, and *Mysostis arvensis*. The use of a mixture of Afalon Dispersive 450 SC and Command 480 EC has shown high efficiency in reducing species, such as *Chenopodium album*, *Raphanus raphanistrum*, *Vicia tetrasperma*, *Cirsium arvense*, *Galeopsis tetrahit*, *Spergula arvensis*, *Veronica hederaefolia*, and *Myosotis arvensis*, while the activity against *Polygoneum lapathifolium*, *Galium aparine*, *Viola arvensis*, and *Stellaria media* is weak.

## 5. Conclusions

The use of covers in the cultivation of very early potato cultivars increased the number and mass of weeds and enriched the species composition of segetal vegetation.

Mechanical and chemical methods proved to be more effective than the mechanical method. The best effectiveness in limiting both fresh and dry weed mass in potato cultivation under cover was achieved using the mechanical and chemical method using a mixture of herbicides Afalon Dispersion 450 SC and Command 480 EC.

Variable courses of atmospheric conditions in the years of the study differentiated both the number of monocotyledonous and dicotyledonous weeds, their fresh and dry mass, and the species composition of segetal vegetation.

**Author Contributions:** Conceptualization, P.P., B.S.; methodology, P.P., B.S., P.B.; software, B.K.-M.; validation, B.S., P.B. and B.K.-M.; formal analysis, P.P.; investigation, P.B., P.P.; data curation, P.P.; writing—original draft preparation P.P., B.S.; writing—review and editing, B.S.; visualization, B.K.-M.; supervision, B.S.; project administration, P.B., P.P.; funding acquisition, P.P. All authors have read and agreed to the published version of the manuscript.

**Funding:** This research received no external funding—own financing.

**Conflicts of Interest:** The authors declare no conflict of interest.

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
