# Peer review of "Biological and Agrotechnical Aspects of Weed Control in the Cultivation of Early Potato Cultivars under Cover"

_agriculture, doi:10.3390/agriculture10090373_

Round 1

Reviewer 1 Report

Dear authors,

Below, you will find detailed comments and suggestions on the manuscript that will help to improve the quality of the paper.

Level of English: Unfortunately, the level of English is moderate and there are several points throughout the manuscript that are not easily understood by the reader. Thus, it is suggested the manuscript  be checked by a native English speaker.

Abstract. It would be preferable to use the phrase “weed management methods” and “mechanical techniques” rather than “chemical care” and “mechanical care” throughout the manuscript. In addition to that it would be better if you try to avoid to use the word object (line 22: …as a standard object…, line 24: ..as a control object…, line 29: in objects covered with covers…in objects without covers…as it sounds too general and it is also somehow confusing for the reader). Please take also into consideration that the abstract should be a total of about 200 words maximum.

Introduction. In this section it is important the current state of the research field to be reviewed carefully and key publications to be cited. Maybe the first paragraph of the Discussion section could be added here in order to provide some additional information about the background of your topic.

Materials and Methods. Please provide more information about the experimental design (which are the main plots, the subplots  and the sub-subplot) and the plot sizes.

In addition, please keep a consistent way of expressing the doses of plant protection products, for example as active ingredient or active substance per ha à a.i./ a.e. ha-1.  Follow the same tactic also in Tables 1 and 2.

Please replace the phrase “care procedures” with weed management practices” (line 65, page 2 as well as line 123, page 4).

Lines 123, 124, page 4: It would be preferable to use the word “quadrat” instead of “frame method”

Line 126, page 4: There is a missing word: The following characteristics were determined… In addition, in this part of Materials an methods it is necessary to provide more information about the assessment of weed infestation and explain how you calculated fresh and air-dry mass (how many plants per weed and also the followed procedure) in order to be more clear for the reader the results of Tables 7 and 8.

Table 3: Please check again the column titles of your table. It is also preferable to use the name of each month.

Line 148, page 5: …dry -<k 0.4; à It seems that something is missing there.

Statistical analysis: Please mention the exact software (SAS/STAT 9.2) that has been used and how did you processed the data (which factors where involved and what was the exact procedure that has been followed).

Results. 

Tables’ headings are not detailed enough so that the reader can understand the data without reading the main text. Please provide a more detailed description (units, definitions, abbreviations). Moreover, different superscript letters (a, b, c, d) should be used along with data to show statistically significant differences. Finally, Table 9 seems to be too complex and is difficult to be understood.

Discussion. The main purpose of this part is discussing the results and how they can be interpreted in perspective of previous studies and of the working hypotheses. So, as it is mentioned above, the first paragraph could be added to the Introduction section. In addition, it is important your manuscript sections be evenly distributed and easy to read, thus it is proposed to maintain similar in size and comprehensive paragraphs throughout this section. Too much and not very well-organized information sometimes can be confusing for the reader.

Conclusions. In this section, one or two well-developed paragraphs summarizing your thoughts and conveying the larger implications of your study they would be more coherent and easier to understand for the reader.

References. The large number of publications in a research article demonstrates thorough research. However, this is a research article and not a review article. Thus, a re-evaluation of the literature is proposed in order to use key references that can add value to the manuscript.

Finally, it would be suggested to include also some correlations among the examined characteristics in order to increase the scientific soundness of the manuscript.

Author Response

The authors of the work would like to thank the Honorable Reviewer for valuable comments and tips that contributed to the improvement of the level of this manuscript

Responses to Review No. 1

1) The summary has been improved, the number of words has been reduced and the nomenclature of weed control methods has been changed, the word "object" has also been removed

2) Moved the first paragraph from the chapter "Discussion" to the chapter "Introduction"

3) The methodology was changed as follows:

  1. a) The size of plots to be harvested is given;
  2. b) Replaced "care methods" with "weed control practices"
  3. c) in Tables 1 and 2 and in the text of the discussion of the results, the doses of herbicides were applied uniformly according to the manufacturer's label;
  4. d) replaced the "frame method" with a "quadrat"
  5. e) the methodology for determining the weed infestation was supplemented;
  6. f) in Table 4, the names of the months have been introduced
  7. g) the missing description was completed in the legend of Table 4
  8. h) in the section "Statistical analysis", the program used was supplemented
  9. i) Statistical calculations were based on a three-way analysis of the variance model and multiple t-Tukey tests at the significance level p = 0.05, which allowed for the correct assessment of the significance of differences.

4) In the chapter "Results" the headings of the tables were corrected

5) The conclusions were shortened and redrafted as suggested by the Reviewer

6) 20 references were removed from the bibliography

7) We intend to carry out and publish the correlations features suggested by the Reviewer in the next work.

Reviewer 2 Report

Dear Authors,

Your manuscript is a thorough piece of work, but I am afraid it needs some reconsideration. Please consult the detailed list I provide below.

Title - The manuscript has a promising title that engages the reader. At the same time, the title seems a bit too generic.

Authors’ list and affiliations - All right.

Abstract - This section is above the maximum word count limit. Please remain under 200 words. Sentence at Line 27 “Studies have shown…” should go into the “Introduction” part of the Abstract. Also, I find that some of the “Methods” part of the Abstract may be omitted, leaving more space for the “Conclusions”, which part now is missing.

Keywords - There are only 5 keywords used (instead of the maximum of 10), and although they fit the subject, they are not specific enough. I suggest the addition of 3-4 more keywords that give the reader more insight into the article, such as “mechanical weed control”, “chemical weed control”, “plastic mulch materials”, “early potato production” or something similar.

Introduction - The Introduction gives enough motivation to the reader, but a little expansion may be needed, as this section is too short. The aim of the study and the test hypothesis is presented, although not clearly. I do not understand what “flat” may mean in Line 52. The main conclusions are missing! Please follow the Instructions to Authors more carefully.

Here https://www.mdpi.com/journal/agriculture/instructions

Materials and Methods - A well-written, thorough section.

Unfortunately, at times, language, term selection becomes a barrier for understanding. In Table 2 for example, what does the term “grace period” refer to? Subsection 1.3 again, is correct, but the Introduction has not given the reason behind the need for this soil survey. The statistical methods used are well described.

Results - How come the results of a soil survey are presented as a result? This subsection (2.1.) should go to M+M to describe the field where the study took place. Results are clearly presented.

Discussion - The first paragraph of the Discussion puts the study into a broader context. Language, term use especially, makes it difficult to follow. The final paragraph (starting at Line 508) does not really give anything plus or extraordinary to the manuscript. It basically states that ‘weed management is efficient when it is done correctly’. I think this paragraph needs some reconsideration.

Conclusions, a separate chapter - It is basically a summary of the findings. It is important that the manuscript has these findings written, but I don’t see the point of having this as a separate section. It does not add anything special to the results that we already know. I think the sentences in this section should be incorporated into the “Results” section.

References - Literature is mostly Polish, meaning that almost two-third of the cited literature is by Polish authors. I would have welcomed if you relied upon a broader, more international spectrum of scientific knowledge.

Inappropriate citation style: In Lines 437 and 441 for example, the cited author’s name and the number are both given, whereas only the number should have been given.

Authors contribution - Fine.

Language - Some terms are difficult to decipher: “object” (instead of treatment or plot(s), perhaps?) “grace period” (withdrawal period, perhaps?), “sheeting” (instead of cover method or mulching, perhaps?). The reader has to de-code these strange terms, and not only it takes time, but the reader may also get easily lost and lose patience.

For example, in the case of “object”, see Line 203 “The objects where ‘Denar’ cv. was grown were characterized by …etc.”. This sentence is weird but starts to make sense if we replace “object” with “plots” or “treatments”.

Also, grammar is not without flaws.

In Lines 2017-209, “The lowest number of weeds in this group was recorded in warm 2016 with optimally distributed rainfall during the growing season. Most of them were recorded in 2017 with…etc.” Although I feel the second sentence refers to “weeds” by “most of them”, but the composition is lazy, because the first sentence has “number of weeds” as a subject, and the subject of the second sentence is only “weeds”.

The same type of mistake is in Lines 151-153 “Optimal water supply to potato plants was observed alternately, with a huge shortage of them in the subsequent month of vegetation.”, where “them” in the second half of the sentence refers to “water supply”, but according to the structure of the sentence, this “them” may also refer to “potato plants”. It is a bit confusing.

What do authors mean by “the plants ripened” as in Line 247?

The Authors may want to consider using expressions and specific terms that are more widely accepted, and/or present a more obvious meaning. I suggest the Authors work on the language to make the text easier to follow.

Line 277 Strange word use “differentiate”

Line 356 Strange reference to percentages “estimated at 10% to 70%.”
Line 357 Wrong form of word “limiting” instead of “limited”.

In the Discussion, I found another common mistake: the way references are introduced. Most of the time it is “in the opinion of” or “reported by”, or “prove”, or “according to”. It is not a serious mistake by any means, but it does wear the reader out. It weakens the text, makes it lifeless. I suggest rephrasing.

I definitely advise the Authors have the text proofread by a native speaker.

Author Response

The authors of the work would like to thank the Honorable Reviewer for valuable comments and tips that contributed to the improvement of the level of this manuscript

Responses to Review No. 2

1) The summary was significantly shortened to the required volume.

2) The keywords have been supplemented.

3) In the chapter "Introduction":

  1. a) the text was supplemented with an explanation of the term covers put on "flat"
  2. b) the first paragraph was moved from the chapter "Discussion";
  3. c) the instructions for the authors have been followed.

4) In the chapter "Material and Methods" the following changes were made:

  1. a) the meaning of the grace period is explained in the legend of table 2;
  2. b) Subchapter 2.1 was moved from "Soil analysis" to the chapter "Material and Methods", as suggested by the Reviewer;
  3. c) the citation method has been corrected as required by the Authors;
  4. d) in lines 151-153 the description of meteorological conditions was reworded;

5) In the Discussion of Results, the last paragraph was deleted, as indicated by the Reviewer,

6) Detailed conclusions, as suggested by the Reviewer, were moved to the chapter "Discussion".

8) 20 references were removed from the bibliography

Reviewer 3 Report

Dear Authors,

The thematic of this manuscript is very actual because weeds are plants that will always grow in arable fields. Hence, ways to limit them should be sought.

Overall, the study was carried out correctly.

The aim, materials and methods, statistical analysis are quite clearly described. The results were interpreted and described correctly. However, there are numerous technical errors and minor inaccuracies in the work that need to be completed or corrected. Changes should be made in References as given below.

Ln 7-13 - to standardize the case of letters in orcid for authors

Ln 9 - correct e-mail (repeated 2 times)

Ln 38, 43, 45…. - without space between literature items

Ln 38 – [1,2,3] change to [1-3]

Ln 43 - [5,6,7,8,9] change to [5-9] - improve throughout the work

Ln 80 - indicate the size of one plot

Ln 89-90 - Table 1 - provide active ingredients of preparations

Ln 95 and Table 2 - Afalon  Dyspersyjny – change to Afalon Dispersive

Table 2 – chlomazon change to clomazone

Ln 96 – 500 g in 1 kg… change to 450 g in 1 kg….

Ln 124 – without …. using the frame method ….

Ln 129 - meteorological conditions to be discussed briefly - synthetically

Ln 137, 142, 157 - word Table 3 - write in the text only once

Ln 143 - provide a shorter title Table 3

Ln 166 - Soil analysis and Table 4 - should be in Material and Methods in Soil sampling and evaluation

Ln 167 – sentence „The field experiments were carried out at the Uhnin Cultivar Test Experimental  Station” – it was in Ln 59-60 

Ln 170-177 - do not specify the values of the features in the text - they are in Table 4

Ln 178-179 - provide a shorter title Table 4

Ln 184 - without "On the other hand"

Ln 185, 192, 198, 202, 210, 218 - write the word Table 5 in the text only once in Ln 183 - do not repeat, this also applies to tables 6-9

Ln 196, 201 - Afalon Dyspersyjny 450 S.C. - change to Afalon Dispersive 450 SC

Ln 220 - without… .approximately…….

Ln 220 - should be changed to…was 20,2 pcs.m-2, on average.

Ln 355-510 - Discussion and Ln 511-536 - Conclusions – it is advisable to shorten - they should be more synthetic

Also:

- you should limit your own citations to 3-4 items

- most of the old references should be eliminated, e.g. 7, 11, 29, 36, 43, 48, 51, 52, 53, 58, 62, 64, 69, 73, 76, 78 ... .. - no quantity is important here but current knowledge

- adapt References to the requirements of the journal (journal names - abbreviations, without ISSN, in Polish, in Polish ... ..)

Author Response

The authors of the work would like to thank the Honorable Reviewer for valuable comments and tips that contributed to the improvement of the level of this manuscript

Responses to Review No. 3

1) Uniform letter size in affiliation, removed double e-mail address

2) In the chapter "Material and Methods" the following changes were made:

  1. a) the citation method has been corrected in accordance with the requirements for the Authors;
  2. b) the size of plots to be harvested is given;
  3. c) the active ingredients of the formulations are listed in Tables 1 and 2
  4. d) subchapter 2.1 with soil analysis was moved to the chapter "Material and Methods",
  5. e) renamed "Afalon Dispersive" to "Afalon Dispersive";

(f) renamed "Chlomazon" to "Clomazone";

  1. g) the error in the content of the active substance of Afalon has been corrected;
  2. h) the "frame method" was replaced with a "quadrat"
  3. i) meteorological conditions, as suggested by the Reviewer, were discussed synthetically;
  4. j) the citation of tables 3 and 6-9 has been limited as suggested by the Reviewer
  5. k) The titles of tables 3 and 4 have been shortened as indicated by the Reviewer;

3) In the "Results" chapter, the data in line 220 was corrected

4) The conclusions were shortened and generalized as suggested by the Reviewer

5) Changes in the bibliography:

  1. a) 20 literature items were removed as indicated by the Reviewer
  2. b) own citations were limited and
  3. c) removed abbreviations: ISSN, ISBN and expression "in polish".
